# MULTITASK CONTRASTIVE LEARNING

## ABSTRACT

Multi-task and contrastive learning are both aimed at enhancing the robustness of learned embeddings. But combining these two fields presents challenges. Supervised contrastive learning brings together examples of the same class while pushing apart examples of different classes, which is intuitive in single-task scenarios. However, contrastive learning becomes less intuitive when dealing with multiple tasks, which might require different notions of similarity. In this work, we introduce a novel method, Multi-Task Contrastive Loss (MTCon), that improves the generalization capabilities of learned embeddings by concurrently incorporating supervision from multiple similarity metrics. MTCon learns task weightings that consider the uncertainty associated with each task, reducing the influence of uncertain tasks. In a series of experiments, we show that these learned weightings enhance out-of-domain generalization to novel tasks. Across three distinct multi-task datasets, we find that networks trained with MTCon consistently outperform networks trained with weighted multi-task cross-entropy in both in-domain and out-of domain multi-task learning scenarios. Code will be made available upon publication.

## 1 INTRODUCTION

Multi-task learning and contrastive learning have each garnered significant attention for their potential to enhance the robustness and generalization capabilities of learned embeddings. Multi-task learning simultaneously solves multiple tasks, exploiting their shared information to produce superior representations and models, particularly when training data is limited (Du et al., 2020; Zhang & Yang, 2021). This approach introduces regularization by compelling the model to excel across diverse tasks, mitigating the risk of overfitting to individual tasks.

Contrastive learning trains embeddings by discriminating similar sample pairs (positive examples) from dissimilar sample pairs (negative examples). Supervised contrastive learning (Khosla et al., 2020) uses examples with the same label as positive examples and different labels for negatives. Self-supervised contrastive learning generates positive pairs by augmenting single examples (Chen et al., 2020; Arora et al., 2019). Embeddings trained with self-supervised and supervised contrastive learning techniques have achieved state-of-the-art performance in a variety of computer vision tasks (Radford et al., 2021; Yuan et al., 2021; Khosla et al., 2020). Given the success of both multi-task and contrastive learning, a natural question arises: can we combine these two fields to improve the generalization of learned embeddings?

Combining multi-task and contrastive learning presents a challenge. The idea underlying supervised contrastive learning, pulling together examples of the same class and pushing apart those of different classes, becomes less straightforward in the context of multi-task learning. Two examples can fall under the same class for one task but fall under different classes for another task. For example, in Figure 1, each of the images of shoes are labeled with category, closure, and gender attributes. Images 1 and 2 are similar in category but are dissimilar in closure and gender, while images 2 and 3 are similar in gender but dissimilar in category and closure. Which images should be pulled together and which should be pushed apart in a contrastive setting? Another challenging factor is that different tasks might have different levels of noise or uncertainty, and incorporating noisy similarity measures can lead to worse, rather than better, generalization performance on new tasks and datasets (Kendall et al., 2018; Mao et al., 2022).

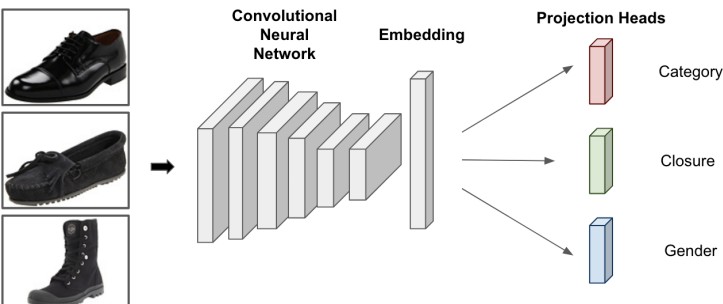

| ID | Category | Closure | Gender |
|----|----------|---------|--------|
| 1 | Shoe | Lace up | Men |
| 2 | Shoe | Slip on | Women |
| 3 | Boot | Lace up | Women |

Figure 1: **Shoe Example.** An example illustrating multiple disjoint similarity relationships between three images of shoes.

In this work, we introduce multi-task contrastive loss (MTCon), a contrastive loss function and architecture that utilizes supervision from multiple tasks and learns to down-weight more uncertain tasks. Our framework is shown in Figure 2. MTCon uses multiple projection heads to learn embeddings based on different metrics of similarity from different tasks. In this way, we are able to represent examples that are positive examples in one projected subspace and negative examples in a different projected subspace. For determining task weighting, we start by understanding the role of task uncertainty on generalization error in the contrastive multi-task setting. We first prove that training on tasks with higher homoscedastic noise or uncertainty can hurt generalization performance in the multi-task contrastive setting. We then construct a weighting scheme that learns to down-weight these uncertain tasks during training. We show through experiments that our weighting scheme allows MTCon to generalize better to unseen tasks.

We evaluate MTCon on three diverse multi-task datasets. We find that networks trained with MTCon consistently outperform networks trained with weighted multi-task cross-entropy by a statistically significant margin of 3.3% in out-of-domain and 1.5% in in-domain multi-task learning scenarios. We also show that embeddings trained with our multi-similarity contrastive loss outperform embeddings trained with traditional self-supervised and supervised contrastive losses and previous multi-similarity contrastive learning methods.

Our main contributions are: 1) We present a novel method, MTCon, for using contrastive learning in the general multi-task setting; 2) We construct a weighting scheme that learns to down-weight uncertain tasks during training and show through experiments that our scheme helps learn embeddings that generalize better to unseen tasks; 3) We empirically demonstrate that networks trained with MTCon out-performs multi-task cross-entropy and previous contrastive methods for both out-of-domain and in-domain tasks.

Figure 2: **Multi-Task Contrastive Network.** Multiple projection heads are trained to learn from multiple metrics of similarity from different tasks. The projection heads are discarded and only the encoding network is kept for downstream tasks. During training, our network is able to learn weightings for each similarity metric based on the task uncertainty.

## 2 RELATED WORK

**Multi-Task Learning.** Multi-task learning aims to simultaneously learn multiple related tasks, and often outperforms learning each task alone (Kendall et al., 2018; Bhattacharjee et al., 2022; Mao et al., 2020; Du et al., 2020). However, if tasks are weighted improperly during training, the performance on some tasks suffer. Various learned task weighting methods have been proposed for

multi-task learning in the vision and language domains (Mao et al., 2022; Kendall et al., 2018; Chen et al., 2018; Sener & Koltun, 2018; Liu et al., 2017; Mao et al., 2020; 2021; Gong et al., 2019). These methods learn task weightings based on task characteristics, and improve the generalization performance towards novel tasks (Mao et al., 2022). This is done by regularizing the task variance using gradient descent (Chen et al., 2018; Mao et al., 2021) or by using adversarial training to divide models into task-specific and generalizable parameters (Liu et al., 2017). Overwhelmingly, these methods are built for multiple tasks trained with likelihood-based losses, such as regression and classification. One of the most popular of these methods models task uncertainty to determine task-specific weighting and automatically learns weights to balance this uncertainty (Kendall et al., 2018). In our work, we adapt automatically learned task weighting to our multi-similarity contrastive loss by predicting similarity uncertainty (Ardeshir & Azizan, 2022).

**Contrastive Representation Learning.** Our work draws from existing literature in contrastive representation learning. Many of the current state-of-the-art vision and language models are trained using contrastive losses (Radford et al., 2021; Yuan et al., 2021; Chen et al., 2020; Khosla et al., 2020; He et al., 2020). Self-supervised contrastive learning methods, such as MoCo and SimCLR, maximize agreement between two different augmentations or views of the same image (He et al., 2020; Chen et al., 2020). Recently, vision-language contrastive learning has allowed dual-encoder models to pretrain with hundreds of millions of image-text pairs (Jia et al., 2021; Radford et al., 2021). The resulting learned embeddings achieve state-of-the-art performance on many vision and language benchmarks (Yuan et al., 2021; Wang et al., 2022; Li et al., 2022). Supervised contrastive learning, SupCon, allows contrastive learning to take advantage of existing labels (Khosla et al., 2020; Yang et al., 2022; Zhang & Yang, 2021). Previously developed conditional similarity networks and similarity condition embedding networks train on multiple similarity conditions by representing these similarities as different triplets (Veit et al., 2017; Tan et al., 2019). Conditional similarity networks learn masks for each metric of similarity; similarity condition embedding networks use an additional conditional weighting branch. Both of these networks optimize a modified form of triplet loss and we compare to both methods in our experiments.

We build on earlier work on the theory underlying the generalization of contrastive based losses. (Arora et al., 2019) analyzes the generalization of InfoNCE loss in the binary case assuming that positive samples are drawn from the same latent classes. Other work studies the behavior of InfoNCE loss from the perspective of alignment and uniformity (Ardeshir & Azizan, 2022; Oh et al., 2018), and shows that generalization error of self-supervised contrastive learning losses can be bounded by the alignment of generated data augmentation strategies (Huang et al., 2021). Other work investigates the generalization error of contrastive learning using label information in order to understand why labeled data help to gain accuracy in same-domain classification tasks (Ji et al., 2021). Though this last work does not present a method that addresses the supervised multi-task contrastive learning problem, we build directly on it to show that training with noisier task labels increases the generalization error bound towards novel tasks for multi-task supervised contrastive loss.

## 3 SETUP AND NOTATION

In this paper, we use $O$ to denote universal constants, and we write $a_k \lesssim b_k$ for two sequences of positive numbers $\{a_k\}$ and $\{b_k\}$ if and only if there exists a universal constant $c > 0$ such that $a_k < c \cdot b_k$ for any $k \geq 0$. Let $|A|$ denote the cardinality of set $A$. We use $|| \cdot ||$ represent the $l_2$ norm of vectors. $\lambda_r(W)$ represents the $r$th eigenvalue of matrix $W$. Let $\mathbb{E}_D[\cdot]$ and $\mathbb{E}_E[\cdot]$ denote the expectation taken with respect to the set of data samples used for training and the data samples with target task labels, respectively.

We assume that during training time, we have access to dataset: $\mathcal{D} = \{x_i, \mathbf{Y}_i\}_i^M$, where $x$ is an image and the $\mathbf{Y}_i = \{y_i^1 ... y_i^C\}$ are distinct categorical attributes associated with the image. We aim to learn an embedding function $f$ that maps $x$ to an embedding space. Let $d$ represent the dimension of the input space and $n = |M|$ represent the number of data samples.

In the typical contrastive training setup, training proceeds by selecting a batch of $N$ randomly sampled data $\{x_i\}_{i=1...N}$. We randomly sample two distinct label preserving augmentations, $\tilde{x}_{2i}$ and $\tilde{x}_{2i-1}$, for each $x_i$ to construct $2N$ augmented samples, $\{\tilde{x}_j\}_{j=1...2N}$. Let $A(i) = \{1, ...2N\}\backslash i$ be the set of all samples and augmentations not including $i$. We define $g$ to be a projection head

that maps the embedding to the similarity space represented as the surface of the unit sphere $\mathbb{S}^e = \{v \in \mathbb{R}^e : ||v||_2 = 1\}$. Finally, we define $v_i = g(h_i)$ as the mapping of $h_i$ to the projection space.

Supervised contrastive learning uses labels to implicitly define the positive sets of examples. Specifically, supervised contrastive learning encourages samples with the same label to have similar embeddings and samples with a different label to have different embeddings. We follow the literature in referring to samples with the same label as an image $x_i$ as the positive samples, and samples with a different label than that of $x_i$'s as the negative samples.

Supervised contrastive learning (SupCon) Khosla et al. (2020) proceeds by minimizing the loss:

$$L^{supcon} = \sum_{i \in I} \frac{-1}{|P(i)|} \sum_{p \in P(i)} \log \frac{\exp(\frac{v_i^\top v_p}{\tau})}{\sum_{a \in A(i)} \exp(\frac{v_i^\top v_a}{\tau})}, \qquad (1)$$

where $|S|$ denotes the cardinality of the set $S$, $P(i)$ denotes the positive set with all other samples with the same label as $x_i$, i.e., $P(i) = \{j \in A(i) : y_j = y_i\}$, $I$ denotes the set of all samples in a particular batch, and $\tau \in \{0, \infty\}$ is a temperature hyperparameter.

## 4 METHODS

In this section, we introduce a novel extension, MTCon, of supervised contrastive learning to the multi-task setting. We start by analyzing the generalization error bound of a simplified version, MTCon-s, highlighting its dependence on the noise/uncertainty in different tasks. Guided by our theoretical findings, we propose a modification of the MTCon-s objective that down-weights uncertain tasks during training to reduce generalization error of the learned embedding to novel tasks.

In contrast to SupCon, our multi-task contrastive approach proceeds by jointly training an embedding space using multiple notions of similarity from different tasks. We do so by training the embedding with multiple projection heads $g^c$ that map the embedding to $C$ projection spaces, where each space distinguishes the image based on a different similarity metric. We define $v_i^c = g^c(h_i)$ to be the mapping of $h_i$ to the projection space by projection head $g^c$. Because each projection space is already normalized, we assume that the each similarity loss is similarly scaled. We define the multi-task contrastive loss to be a summation of the supervised contrastive loss over all conditions $L^{mtcon^s} = \sum_{c \in C, i \in I} L_{c,i}^{mtcon^s}$ where each conditional $L_{c,i}^{mtcon-s}$ is defined as in equation 2. Specifically,

$$L_{c,i}^{mtcon^s} = \frac{-1}{|P^c(i)|} \sum_{p \in P^c(i)} \log \frac{\exp(\frac{v_i^{c\top} v_p^c}{\tau})}{\sum_{a \in A(i)} \exp(\frac{v_i^{c\top} v_a^c}{\tau})}, \qquad (2)$$

where $P^c(i)$ is defined as the positive set under similarity $c$ such that for all $j \in P^c(i)$, $y_j^c = y_i^c$.

### 4.1 EFFECTS OF TASK UNCERTAINTY ON GENERALIZATION ERROR

Our novel MTCon-s objective function has the advantage of leveraging different notions of similarity in learning the emeddings. However, in the presence of highly un-informative (i.e., high noise) tasks, the MTcon-s objective might have poor generalization. In this section, we present a formal argument that shows how the generalization error for downstream tasks depends on the noise of the source tasks. We extend previous work (Ji et al., 2021; Bai & Yao, 2012) by postulating that the input data (i.e. $x$) for each task $t$ is a Gaussian mixture model with $r + 1$ components shared across the $T$ tasks. However, the mixture probabilities $p_{k,t}$, and the noise level $\xi^{k,t}$ vary across tasks. Specifically, we assume that the data is generated under the spiked covariance model with homoscedastic noise under the multi-task setting:

$$x^{k,t} = \mu^k + \xi^{k,t}, \ \ \text{Cov}(\xi^{k,t}) \sim N(0, \Sigma^{k,t}), \forall k \in [r+1], t \in [T], \text{ and } x^t = \sum_{k}^{r+1} p_{k,t} x^{k,t} \qquad (3)$$

Specifically, we make the assumption that $\Sigma^{k,t} = \sigma_t^2 \cdot I$ for all $k \in \{1, ..., r+1\}$ where $\sigma_t^2$ represents the noise variance parameter dependent on each task $t$. Following (Ji et al., 2021), we make the additional assumptions that the covariance matrix $\Sigma^{k,t}$ satisfies the regular covariance condition, that the covariance of the noise is of the same order as the features, and that the feature matrix satisfies the incoherence condition as defined in the Appendix. Full statements for all of our assumptions are presented in the Appendix. We aim to learn feature embeddings that generalize to target tasks, recovering the orthonormal basis $W^*$ of the span of $\mu^k$. Under the given assumptions, recovering the span of $\mu^k$ allows us to learn a representation that will cover the features necessary to learn a linear transformation to the target task.

Under these assumptions, we analyze the downstream performance of linear representations and simple predictors, which take a linear transformation of the representation as an input. Specifically, for a representation matrix $W$, and a set of weights $w$, $f_{w,W}(x) = w^\top W x$. Note that effectively, $Wx_i = h_i$ from the last section. For simplicity, we focus on the mean squared error $\ell(f_{w,W}(x), y) = (w^\top W x - y)^2$, but we note that our analysis is extendable to other losses.

**Theorem 1** *Suppose $n > d \gg r$, $T > r$ and $\lambda_{(r)}(\sum_{t=1}^T w_t w_t^\top) > c$ for some constant $c > 0$. Let $W_{CL}$ be the learned representation using MTCon-s and $W^*$ be the optimal true representation. Then, the prediction risk of the downstream task can be bounded as:*

$$\mathbb{E}_D\big[\inf_{w \in \mathbb{R}^r} \mathbb{E}_E[\ell(f_{w,W_{CL}}(x), y)] - \inf_{w \in \mathbb{R}^r} \mathbb{E}_E[\ell(f_{w,W^*}(x), y)] \lesssim \sqrt{\frac{dr}{n}}(\sum_{t=1}^T \sigma_t)$$

The proof is presented in the Appendix. Theorem 1 shows that the generalization error for downstream tasks depends on the sum of $\sigma_t$, the task-specific variances over the noise variable. This in turn implies that the generalization error deteriorates if the source data includes noisy tasks.

## 4.2 CONTRASTIVE TASK WEIGHTING SCHEME

In the simplified formulation of our multi-similarity contrastive loss function, each similarity is weighted equally. However, as shown in the previous section, tasks with higher homoscedastic noise or uncertainty can hurt generalization performance in the multi-task contrastive setting. Previous work in general multi-task learning has suggested using irreducible uncertainty of task predictions in a weighting scheme (Kendall et al., 2018). For example, tasks where predictions are more uncertain are weighted lower because they are less informative.

Such notions of uncertainty are typically predicated on an assumed parametric likelihood of a label given inputs. However, this work is not easily adapted to multi-similarity contrastive learning because 1) contrastive training does not directly predict downstream task performance and 2) the confidence in different similarity metrics has never been considered in this setting. In contrastive learning, the estimate of interest is a similarity metric between different examples rather than a predicted label, so downstream task performance is not directly predicted by training results. Furthermore, previous work in contrastive learning has only focused on modeling data-dependent uncertainty, or how similar a sample is to negative examples within the same similarity metric. To our knowledge, we are the first to utilize uncertainty in the training tasks and their corresponding similarity metrics as a basis for constructing a weighting scheme for multi-similarity contrastive losses.

We do this in two ways: 1) we construct a pseudo-likelihood function approximating task performance and 2) we introduce a similarity dependent temperature parameter to model relative confidence between different similarity metrics. We present an extension to the contrastive learning paradigm that enables estimation of the uncertainty in similarity metrics. Our estimate of uncertainty enables us to weight the different notions of similarity such that noisy notions of similarity are weighted lower than more reliable notions.

Our approach proceeds by constructing a pseudo-likelihood function that approximates task performance. We show in the Appendix that maximizing our pseudo-likelihood also maximizes our MTCon objective function. This pseudo-likelihood endows the approach with a well-defined notion of uncertainty that can then be used to weight the different similarities.

Let $v_i^c$ be the model projection head output for similarity $c$ for input $x_i$. Let $\mathbf{Y}^c$ be the $c$th column in $\mathbf{Y}$. We define $P_y^c = \{x_j \in \mathcal{D} : \mathbf{Y}_j^c = y\}$ to be the positive set for label $y$ under similarity metric

$c$. We define the classification probability $p(y|v_i^c, D, \tau)$ as the average distance of the representation $v_i^c$ from all representations for inputs conditioned on the similarity metric. Instead of directly optimizing equation 1, we can maximize the following pseudo-likelihood:

$$p(y|v_i^c, D, \tau) \propto \frac{1}{|P_y^c|} \sum_{p \in P_y^c} \exp(\frac{v_i^{cT} v_p^c}{\tau}). \tag{4}$$

Note that optimizing 4 is equivalent to optimizing 1 by applying Jensen's inequality. By virtue of being a pseudo-likelihood, equation 4 provides us with a well-defined probability associated with downstream task performance that we can use to weight the different tasks. We will next outline how to construct this uncertainty from the pseudo-likelihood defined in equation 4.

We assume that $v^c$ is a sufficient statistic for $y^c$, meaning that $y^i$ is independent of all other variables conditional on $v^i$. Such an assumption reflects the notion that $v^c$ is an accurate estimation for $y^c$. Under this assumption the pseudo-likelihood expressed in 4 factorizes as:

$$p(y^1, ...y^C | v_i^1, ...v_i^C, D, \tau) = p(y^1 | v_i^1, D, \tau)...p(y^C | v_i^C, D, \tau). \tag{5}$$

Previous work in contrastive learning modifies the temperature to learn from particularly difficult data examples Zhang et al. (2021); Robinson et al. (2020). Inspired by this, we adapt the contrastive likelihood to incorporate a similarity-dependent scaled version of the temperature. We introduce a parameter $\sigma_c^2$ for each similarity metric controlling the scaling of temperature and representing the similarity dependent uncertainty in Equation 6.

$$p(y|v_i^c, D, \tau, \sigma_c^2) \propto \frac{1}{|P_y^c|} \sum_{p \in P_y^c} \exp(\frac{v_i^{cT} v_p^c}{\tau \sigma_c^2}) \tag{6}$$

The negative log-likelihood for this contrastive likelihood can be expressed as Equation 7.

$$-\log p(y|v_i^c, D, \tau, \sigma_c^2) \propto \frac{1}{\sigma_c^2} \sum_{i=I} L_{c,i}^{mtcon^s} + 2\log(\sigma_c) \tag{7}$$

Extending this analysis to consider multiple similarity metrics, we can adapt the optimization objective to learn weightings for each similarity as in Equation 8.

$$\text{argmin}_{f, g_1, ...g_C, \sigma_1, ...\sigma_C} (\sum_{c \in C} (\frac{1}{\sigma_c^2} \sum_{i=I} L_{c,i}^{mtcon^s} + 2\log(\sigma_c))) \tag{8}$$

During training, we learn the $\sigma_c$ weighting parameters through gradient descent. After learning the weighting parameters $\sigma_c$, we can define the weighted loss function as $L^{mtcon} = \sum_{c \in C} (\frac{1}{\sigma_c^2} \sum_{i=I} L_{c,i}^{mtcon^s} + 2\log(\sigma_c))$.

## 5 EXPERIMENTS

We first evaluate the robustness of our learned embeddings to task uncertainty. We show that when we introduce task noise, 1) MTCon learns to down-weight noisy tasks, and 2) the resulting learned embeddings generalize better to novel tasks. We then show that the generalization performance of embeddings trained with MTCon is superior to that of embeddings trained with multi-task cross-entropy or with previous multi-task contrastive losses. Finally, we show that even on in-domain tasks networks trained with our multi-similarity contrastive loss significantly outperform networks trained with existing self-supervised, single-task supervised, and previous multi-task contrastive losses.

**Datasets.** We use three multi-task datasets: Zappos50k (Yu & Grauman, 2014; 2017), MEDIC (Alam et al., 2022; 2018; 2020; Mouzannar et al., 2018; Nguyen et al., 2017), and CUB200-2011 (Wah et al., 2011). Zappos50k contains 50,000 $136 \times 102$ images of shoes. We train models on three tasks: the category of shoe, the suggested gender of the shoe, and the closing mechanism of the shoe. We use the brand of the shoe for the out-of-domain task. We use the published splits and resize all images to $112 \times 112$. MEDIC contains $\sim 71,000$ images of disasters collected from social media. The dataset includes four disaster-related tasks that are relevant for humanitarian aid: the disaster type, the informativeness of the image for humanitarian response, categories relevant to

humanitarian response, and the severity of the damage of the event. For the out-of-domain analysis, we hold out each task from training and then attempt to predict the held-out task during evaluation. We use the published splits. All images are resized to $224 \times 224$. CUB200-2011 has 11,788 labeled images of 200 different bird species. We train models on three tasks: the size, the shape, and the primary color of the bird. We evaluate on species classification for the out-of-domain task. We use the published train/test split and separate $10\%$ of the training set as a validation set. All images are resized to $224 \times 224$.

**Implementation.** Consistent with previous work (Chen et al., 2020; Khosla et al., 2020; Huang et al., 2021), images are augmented by applying various transformations to increase dataset diversity. We train using standard data augmentations: random crops, flips, and color jitters. Zappos50k encoders use ResNet18 backbones with projection heads of size 32. CUB200-2011 and MEDIC encoders use ResNet50 backbones with projection spaces of size 64 (He et al., 2016). All models are pretrained on ImageNet (Deng et al., 2009). All networks are trained using an SGD with momentum optimizer for 200 epochs with a batch-size of 64 and a learning rate of 0.05, unless otherwise specified. We use a temperature of $\tau = 0.1$. To evaluate the quality of the learned encoder, we train a linear classifier for 20 epochs and evaluate top-1 accuracy. Standard deviations are computed by bootstrapping the test set 1000 times.

**Baselines.** We compare MTCon with multi-task, single-task, and self-supervised baselines:

**(1) Multi-Task Cross-Entropy (XEnt MT)** We train a weighted multitask cross-entropy network with all available tasks (Kendall et al., 2018). We train each network with a learning rate of 0.01 for 200 epochs. **(2) Conditional Similarity Network (CSN)** Following the procedure in (Veit et al., 2017), we train a conditional similarity network that learns the convolutional filters, embedding, and mask parameters together. 10,000 triplets are constructed from all the similarities available in each training dataset. **(3) Similarity Condition Embedding Network (SCE-Net)** Following (Tan et al., 2019), we train a SCE-Net for each dataset treating each training task as a similarity condition. The same training triplets are used as for the CSN networks. **(4) Single-Task Cross-Entropy (XEnt)** We train single-task cross-entropy networks for each training task with a learning rate of 0.01 for 200 epochs. **(5) SimCLR and SupCon Networks** We train a SimCLR network for each dataset and individual SupCon networks with each of the similarity metrics represented in the training dataset. We pretrain with a temperature of 0.1 for all contrastive networks, which is the typical temperature used for SimCLR and SupCon (Chen et al., 2020; Khosla et al., 2020). For evaluation, we fine-tune a classification layer on the frozen embedding space.

**MTCon Weighting Improves Robustness to Task Uncertainty.** We first evaluate the responsiveness of our learned embeddings to similarity uncertainty. Since the true level of task noise (similarity metric uncertainty) is unobserved, we use a semi-simulated approach, where we simulate uncertain similarities in both the Zappos50k and MEDIC datasets.

For the Zappos50k dataset, we train the encoder using the category, closure, and gender tasks. To introduce task uncertainty, we randomly corrupt the closure task by corruption proportion $\rho$. We randomly sample $\rho$ of the closure labels, and randomly reassign the label amongst all possible labels. Note that when $\rho = 1.0$, all labels are randomly sampled equally from the available closure labels. When $\rho = 0.0$, all labels are identical to the original dataset. For the MEDIC dataset, we train the encoder using the disaster types, humanitarian, and informative similarity metrics. We corrupt the disaster type task to introduce task uncertainty.

For the Zappos50k dataset, we evaluate the top-1 classification accuracy on an out-of-domain task, brand classification, and on an in-domain task, the corrupted closure classification. Similarly, for the MEDIC dataset, we evaluate the top-1 classification accuracy on an out-of-domain task, damage-severity classification, and on an in-domain task, the corrupted disaster-type classification.

Figure 3 shows the results from this analysis. The top panel shows that, as expected, as $\rho$ increases, MTCon learns to down-weight the noisy task. The middle and bottom panels show how out-of-domain and in-domain evaluation accuracy changes as we change task uncertainty. As expected, as $\rho$ increases to 1, the in-domain classification accuracy for both the equal-weighted and weighted MTCon learned embeddings decreases to random. The out-of-domain classification accuracy for the weighted MTCon learned embeddings is more robust to changes in $\rho$ than the unweighted MTCon

learned embeddings. This is because the weighted version of MTCon automatically learns to down-weight uncertain tasks during encoder training.

Figure 3: **MTCon downweights uncertain tasks to improve generalizability to out-of-domain classification.** Unweighted and weighted versions of MTCon are trained on increasing task corruption. The x-axis on all plots represents the amount of task corruption $\rho$. The top row shows that weighted MTCon learns to downweight the corrupted task. The middle row shows that there is no meaningful difference in performance on the corrupted task. The bottom row shows that weighted MTCon generalizes better to out-of-domain tasks than unweighted MTCon.

**Generalization Performance.** We compare the out-of-domain performance of MTCon against multi-task cross-entropy and previous contrastive multi-similarity methods for out-of-domain tasks on the Zappos50k, MEDIC, and CUB200-2011 datasets. We find that MTCon outperforms other multi-task trained methods on out-of-domain tasks for all datasets. On average across datasets, MTCon improves upon multi-task cross-entropy by $3.3\%$, and improves performance for all tasks except for MEDIC informativeness. The informativeness task seems to be carry little information about the other tasks, as evidenced by the fact that, as shown elsewhere, including it hurts performance for other tasks (Alam et al., 2022).

Table 1: **Out-of-domain Performance.** Out-of-domain classification accuracy on hold-out tasks across three multi-task datasets for multi-task learning methods.

| Loss | Zappos50k | CUB200-2011 | MEDIC | | | |
|---|---|---|---|---|---|---|
| | Brand | Species | Severity | Type | Human | Inform. |
| XEnt MT | 32.10 (1.48) | 41.23 (0.47) | 79.51 (0.36) | 75.02 (0.38) | 79.77 (0.4) | **86.18 (0.3)** |
| CSN | 25.72 (2.03) | 34.15 (0.51) | 66.71 (0.35) | 65.18 (0.36) | 67.22 (0.38) | 75.56 (0.3) |
| SCE-Net | 28.72 (1.79) | 38.91 (0.45) | 69.92 (0.31) | 67.27 (0.33) | 71.23 (0.31) | 78.62 (0.31) |
| MTCon | **42.62 (1.52)** | **43.07 (0.48)** | **80.98 (0.32)** | **76.17 (0.32)** | **81.45 (0.34)** | 85.22 (0.3) |

**In-domain Classification Performance.** To evaluate the quality of the learned embedding spaces, we measure classification accuracy on all training tasks for Zappos50k, MEDIC, and CUB200-2011. We report the average accuracy and the standard deviation for all tasks. For the Zappos50k and CUB200-2011 datasets, Table 2, MTCon has the highest classification accuracy of the models. For MEDIC, Table 3, MTCon out performs all of the contrastive learning techniques on all tasks.

However, for three of the MEDIC tasks, the best performance is achieved by one of cross-entropy methods (but different methods dominate for different tasks). We hypothesize that this may be related to the inherent uncertainty of some of the tasks, as observed in Alam et al. (2018; 2022). For all datasets, CSN and SCE-Net achieve accuracies that are lower than the single-task supervised networks. We believe this is because conditional similarity loss is trained with triplet loss Hoffer & Ailon (2015), which others have shown performs less well than N-pairs loss and supervised contrastive learning for single-task learning Sohn (2016); Khosla et al. (2020). More qualitative analysis of the learned similarity subspaces (i.e., TSNE visualizations) is in the Appendix.

Table 2: **In-domain Performance for Zappos50k and CUB200-2011.** MTCon outperforms all baselines on training tasks. Note that the entries for XEnt and SupCon represent separately trained supervised models for each task.

| | Zappos50k | | | CUB200-2011 | | |
|---|---|---|---|---|---|---|
| **Loss** | Category | Closure | Gender | Shape | Size | Primary Color |
| XEnt | 96.64 (0.34) | 92.28 (0.35) | 83.09 (0.60) | 55.76 (0.50) | 55.91 (0.48) | 32.61 (0.45) |
| XEnt MT | 96.98 (0.29) | 93.33 (0.36) | 85.07 (0.55) | 54.87 (0.49) | 56.96 (0.47) | 33.18 (0.45) |
| SimCLR | 90.05 (0.43) | 81.30 (0.49) | 69.10 (0.84) | 34.20 (0.46) | 52.43 (0.48) | 28.51 (0.43) |
| SupCon | 96.95 (0.29) | 91.75 (0.41) | 85.11 (0.58) | 55.92 (0.49) | 58.13 (0.48) | 33.28 (0.47) |
| CSN | 83.33 (0.32) | 72.12 (0.36) | 69.21 (0.60) | 45.14 (0.49) | 48.24 (0.45) | 25.23 (0.42) |
| SCE-Net | 86.23 (0.31) | 75.32 (0.33) | 71.32 (0.59) | 48.29 (0.41) | 51.53 (0.44) | 28.78 (0.41) |
| MTCon | **97.17 (0.27)** | **94.37 (0.35)** | **85.98 (0.56)** | **56.88 (0.49)** | **59.32 (0.48)** | **35.97 (0.45)** |

Table 3: **In-domain Performance for MEDIC.** MTCon outperforms all contrastive learning baselines on training tasks. Note that the entries for XEnt and SupCon represent separately trained supervised models for each task.

| | MEDIC | | | |
|---|---|---|---|---|
| **Loss** | Damage severity | Disaster types | Humanitarian | Informative |
| XEnt | **81.39 (0.35)** | 78.98 (0.35) | **82.1 (0.37)** | 85.68 (0.3) |
| XEnt MT | 81.01 (0.36) | 78.04 (0.32) | **82.25 (0.35)** | 86.01 (0.29) |
| SimCLR | 74.9 (0.4) | 68.5 (0.42) | 73.89 (0.4) | 78.67 (0.33) |
| SupCon | 80.26 (0.33) | 78.33 (0.37) | 74.89 (0.39) | 84.02 (0.3) |
| CSN | 75.13 (0.4) | 70.02 (0.37) | 70.52 (0.38) | 76.28 (0.32) |
| SCE-Net | 77.25 (0.42) | 71.15 (0.39) | 72.12 (0.42) | 77.52 (0.33) |
| MTCon | 81.0 (0.3) | **79.14 (0.31)** | 81.69 (0.3) | 85.15 (0.3) |

## 6 CONCLUSION

In this work, we introduce a method for learning representations using multi-task contrastive loss (MTCon). MTCon uses multiple projection heads to represent examples that may belong to the same class under one task but to different classes under another task. It uses a task uncertainty-based weighting scheme that down-weights uncertain tasks to improve generalization to novel downstream tasks. In a set of experiments, we demonstrate that our MTCon learned embeddings generalize better than embeddings trained with previous multi-task baselines to novel tasks.

A limitation is that our mathematical analysis of the impact of task noise on the multi-task contrastive learning generalization error makes simplifying assumptions that may not hold in practice, including the assumption that source tasks are abundant enough to recover core features necessary for the target task. However, our results on three multi-task datasets show that MTCon works well in practice to train models that generalize to novel tasks. Another limitation of MTCon is that we assume that there exists task-specific noise that we can learn. Our experiments indicate that this assumption holds to varying degrees in different tasks.

In conclusion, we show that we can combine multi-task and contrastive learning to build models that generalize well to novel tasks.

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

# A    APPENDIX

We provide detailed computations and additional experimental results in the Appendix.

## A.1    PROOF OF THEOREM 1

**Proof Strategy**    Our proof follows the proof of Theorem 4.7 in (Ji et al., 2021). We first provide a bound on the sine distance between the orthonormal bases of optimal $W^*$ and learned $W_{CL}$, $U^*$ and $U_{CL}$, as estimated by our proposed simplified contrastive loss MTCon-s. We then provide a bound on the downstream task generalization error as a function of the bound on the sine difference between $U^*$ and $U_{CL}$.

**Notation**    First, we provide some additional notation. We use $\Theta$ to denote universal constants. We use $||\cdot||_2$ and $||\cdot||_F$ to represent the spectral norm of matrices and the Frobenius norm of matrices respectively. Let $\mathbb{O}_{d,r}$ be a set of $d \times r$ orthogonal matrices. $\lambda_r(W)$ represents the $r$th eigenvalue of matrix $W$. We use $||\sin\Theta(U_1, U_2)||_F$ to refer to the sine distance between two orthogonal matrices $U_1, U_2 \in \mathbb{O}_{d,r}$, defined by $||\sin\Theta(U_1, U_2)||_R := ||U_{1\perp}^T U_2||_F$. We additionally define $\{e_i\}_{i=1}^d$ to denote the canonical basis in $d$-dimensional Euclidean space $\mathbb{R}^d$, where $e_i$ is the vector whose $i$-th coordinate is 1 and all other coordinates are 0. We define $\mathbf{1}\{A\}$ to be the indicator function that takes 1 when $A$ is true and 0 otherwise. Let $\Delta(M)$ represent the matrix $M$ with diagonal entries set to 0. We provide some additional definitions below:

**Definition 1** *We define the incoherence constant of $U \in \mathbb{O}_{d,r}$ as*

$$I(U) = \max_{i \in [d]} ||e_i^T U||^2$$

*Intuitively, this constant measures the similarity between $U$ and the canonical basis.*

**Assumptions**    Here we present the standard spiked covariance model assumptions for the multi-task setting and the three assumptions necessary for our theorem. Under the standard spiked covariance model, we assume that $||\mu^k|| = \sqrt{r}\nu, \forall k \in [r+1]$ where $\nu$ represents the scaling factor of the covariance matrix. We assume that our samples are drawn from $r + 1$ different classes for each task $t$, with probability $p_{k,t}$ for each class $k \in [r+1]$ and $\sum_{k=1}^{r+1} p_{k,t} = 1$. To ensure identifiability since the multi-class problem is invariant under translation, we additionally assume $\sum_{k=1}^{r+1} p_k \mu_k = 0$.

Let $\Lambda_t = \sum_{k=1}^{k+1} p_{k,t} \mu_k \mu_k^T$. We additionally assume $\text{rank}(\Lambda_t) = r$ and $C_1\nu < \lambda_{(r)}(\Lambda_t) < \lambda_{(1)}(\Lambda_t) < C_2\nu$ for constants $C_1$ and $C_2$. Finally, we assume that tasks are abundant enough to recover features completely via the labeled data.

Following Assumptions 3.4-3.6 in (Ji et al., 2021), we make the following assumptions under the multi-task spiked covariance model.

**Regular Covariance Condition.**    The condition number of the covariance matrix, $\Sigma_{k,t}$, satisfies $\kappa := \sigma_{1,t}^2 / \sigma_{r+1,t}^2 < C$, where $\sigma_{j,t}^2$ represents the $j$th largest number among $\sigma_{1,t}^2 ... \sigma_{r+1,t}^2$.

**Signal to Noise Ratio Condition** We define the task noise ratio $\rho_t := \nu/\sigma_t$. We assume that $\rho_t = \Theta(1)$ or that the covariance of the noise is the same order as that of the core features for each task.

**Incoherent Condition** The incoherent constant of the feature matrix $W^* \in \mathbb{O}_{d,r}$ satisfies $I(W^*) = O(r \log d/d)$

**Definition** Our proof will rely on an analysis of a hybrid loss that combines the self-supervised loss with supervised contrastive losses (SupCon) with arbitrary weights for each task.

We first must define the matrix loss for self-supervised and supervised contrastive loss. Given two augmentation functions $a_1, a_2$ and $n$ training samples, the augmented views for example $x_i$ are given by $\{(a_1(x_i), a_2(x_i)\}$ and the corresponding positive and negative samples are defined by $\{a_s(x_i) : s \in [2] \backslash \{v\}\}$ and $\{a_s(x_j) : s \in [2], j \in [n] \backslash \{i\}\}$ respectively. Then, the self-supervised contrastive learning loss can be written as:

$$L_{\text{SelfCon}}(W) = -\frac{1}{2n} \sum_{i=1}^{n} \sum_{v=1}^{2} [\langle W a_v(x_i), W a_{[2] \backslash \{v\}}(x_i) \rangle - \sum_{j \neq i} \sum_{s=1}^{2} \frac{\langle W a_v(x_i), W a_s(x_j) \rangle}{2n-2}]$$
$$+ \frac{\lambda}{2} ||WW^\top||_F^2$$

Under the supervised setting, for the $K$-class classification problem for task $t$, given $n_{k,t}$ samples for each class $k \in [K] : \{x_i^k : i \in [n_{k,t}]\}_{k=1}^{K}$ with $n := \sum_{k=1}^{K} n_{k,t}$, we define the positive and negative samples for $x_i^{k,t}$ as $\{x_j^{k,t} : j \in [n_{k,t}] \backslash i\}$ and $\{x_j^{s,t} : s \in [K] \backslash k, j \in [n_{s,t}]\}$ respectively. The supervised contrastive learning loss for the weight matrix for a single task can be written as:

$$L_{\text{SupCon}}(W, t) = -\frac{1}{nK} \sum_{k=1}^{K} \sum_{i=1}^{n} [\sum_{j \neq i} \frac{\langle W x_i^{k,t}, W x_j^{k,t} \rangle}{n-1} - \sum_{j=1}^{n} \sum_{s \neq k} \frac{\langle W x_i^{k,t}, W x_j^{s,t} \rangle}{n(K-1)}] + \frac{\lambda}{2} ||WW^\top||_F^2$$

We consider optimization under the hybrid contrastive loss:

$$\min_{W \in \mathbb{R}^{r \times d}} L(W) := \min_{W \in \mathbb{R}^{r \times d}} L_{\text{SelfCon}}(W) + \sum_{t=1}^{T} \alpha_i L_{\text{SupCon},t}(W)$$

Note that the setting under which $\alpha_t \to \infty$ at the same rate for the hybrid loss exactly matches our simplified multi-task contrastive loss. Also note that MT-Con is the same as the second term in the previous equation with $\alpha_i = \frac{1}{\sigma_i^2}$. Next, we provide some restatements of Lemmas in (Ji et al., 2021) to help with our proof.

Let $H = I_n - \frac{1}{n} 1_n 1_n^\top$. Define matrices $\hat{M}$ and $\hat{N}_i$ as follows, $\hat{M} := \frac{1}{n}(\Delta(XX^\top) - \frac{1}{n-1} X(1_n 1_n^\top - I_n)X^\top)$ and $\hat{N}_t := \frac{1}{(n-1)^2} X_t H y_t y_t^\top H X_t^\top$. Let $M = [\mu_1, ... \mu_k]$ represent the target component means. Intuitively, $\hat{M}$ represent the learned component means and $\hat{N}_t$ the data centering matrix.

**Lemma 1 (Restated Eq. 54 in (Ji et al., 2021))** *Let the bound for the 2 norm expectation of matrix $\hat{M}_2 - M$ which represents learned and target component means:*

$$\mathbb{E}||\hat{M} - M||_2 \lesssim \nu^2 (\frac{r}{d} \log d + \sqrt{\frac{r}{n}} + \frac{r}{n}) + \sigma_1^2(\sqrt{\frac{d}{n}} + \frac{d}{n}) + \sigma_1 \nu \sqrt{\frac{d}{n}}$$

**Lemma 2 (Restated Lemma C3 in (Ji et al., 2021))** *Let $U^*$ represent the orthonormal basis of learned representation $W$. Let the task label be generated by $y = \langle w^*, z \rangle$. Under the assumptions and conditions for a single task, we can find an event $A$ such that $\mathbb{P}(A^C) = O(\sqrt{d/n})$ and:*

$$\mathbb{E}[|| \frac{1}{(n-1)^2} XHyy^\top HX^\top - \nu^2 U^* w^* w^{*\top} U^{*\top}||_F \mathbf{1}\{A\}] \lesssim \sqrt{\frac{d}{n}} \sigma \nu$$

**Lemma 3 (Restated Eq. 92 in (Ji et al., 2021))** *Let the target matrix for all tasks be* $N = \nu^2 U^* U^{*\top} + \sum_{t=1}^{T} \alpha_t \nu^2 U^* w_t w_t^\top U^{*\top}$. *Define* $\sigma_1 = \max_{t \in T}\{\sigma_t\}$. *The upper bound between N and* $\hat{N}$ *can be represented:*

$$\mathbb{E}||\hat{N} - N||_2 1\{\cap_{i=1}^{T} A_i\} \leq \frac{1}{4}\mathbb{E}||\hat{M}_2 - M||_2 + \sum_{t=1}^{T} \alpha_t \mathbb{E}||\hat{N}_t - \nu^2 U^* w_t w_t^\top U^{*\top}||_F \mathbf{1}\{A_t\}$$

**Lemma 4 (Restated Theorem 4.1 in (Ji et al., 2021))** *Under the assumptions with sample size* $m$, *let* $W_{CL}$ *be any solution that minimizes the hybrid contrastive loss. Denote its singular value decomposition as* $W_{CL} = (U_{CL}\Sigma_{CL}V_{CL}^\top)^\top$, *then*

$$\mathbb{E}||\sin\Theta(U_{CL}, U^*)||_F \lesssim \frac{1}{1+\alpha}(\frac{r^{3/2}}{d}\log d + \sqrt{\frac{dr}{n}}) + \frac{\alpha}{1+\alpha}\sqrt{\frac{dr}{m}}$$

**Lemma 5 (Restated Lemma B.22 in (Ji et al., 2021))** *For any* $U \in \mathbb{O}_{d,r}$,

$$\inf_{w \in \mathbb{R}^r}\mathbb{E}_E[\ell(f_{w,W_{CL}}(x), y)] - \inf_{w \in \mathbb{R}^r}\mathbb{E}_E[\ell(f_{w,W^*}(x), y)] = O((1+\rho^{-1})\mathbb{E}_D[|||sin\Theta(U, U^*)||_2||w^*||^2)$$

Combining the computed bound with Lemma 4 gives us the desired bound

$$\mathbb{E}_D[inf_{w \in \mathbb{R}^r}\mathbb{E}_E[\ell(f_{w,W_{CL}}(x), y)] - \inf_{w \in \mathbb{R}^r}\mathbb{E}_E[\ell(f_{w,W^*}(x), y)]] \lesssim \sqrt{\frac{dr}{n}}(\sum_{i=1}^{T} \sigma_t)$$

**Theorem A 1** *Under the multi-task supervised contrastive learning setting, we can compute the upper bound between the contrastive learned and optimal orthonormal basis of the representations,*

$$\mathbb{E}||sin(\Theta(U_{CL}, U*))||_F \lesssim \sqrt{\frac{dr}{n}}(\sum_{t=1}^{T} \sigma_t)$$

**Proof:** We derive an upper bound on the sine distance between the contrastive learned and optimal orthonormal basis. Optimizing the hybrid loss is equivalent to finding the top-r eigenspace of the matrix

$$\frac{1}{4n}(\Delta(XX^T) - \frac{1}{n-1}X(1_n 1_n^T - I_n)X^T) + \sum_{i=1}^{T}\frac{\alpha_i}{(n_i-1)^2}X_i H_{n_i}y_i y_i^T H_{n_i}X_i^T$$

[Eq. 91 in (Ji et al., 2021)]

Again, let $H = I_n - \frac{1}{n}1_n 1_n^\top$. Define matrices $\hat{M}$ and $\hat{N}_i$ as follows, $\hat{M} := \frac{1}{n}(\Delta(XX^\top) - \frac{1}{n-1}X(1_n 1_n^\top - I_n)X^\top)$ and $\hat{N}_t := \frac{1}{(n-1)^2}X_t H y_t y_t^\top H X_t^\top$. Let $M = [\mu_1,...\mu_k]$ represent the target component means. Intuitively, $\hat{M}$ represent the learned component means and $\hat{N}_t$ the data centering matrix. We will proceed by analyzing each of these two terms separately.

From Lemma 1, we can bound the 2-norm expectation of matrix $\hat{M} - M$ as follows,

$$\mathbb{E}||\hat{M} - M||_2 \lesssim \nu^2(\frac{r}{d}\log d + \sqrt{\frac{r}{n}} + \frac{r}{n}) + \sigma_1^2(\sqrt{\frac{d}{n}} + \frac{d}{n}) + \sigma_1 \nu\sqrt{\frac{d}{n}}$$

Next we analyze the second term in the expression. By Lemma 2, for the multi-task setting, for each task $t$, we can find an event $A_t$ such that $\mathbb{P}(A_t^C) = O(\sqrt{d/n})$ and:

$$\mathbb{E}[||\hat{N}_t - \nu^2 U^* w_t^* w_t^{*\top} U^{*\top}||_F \mathbf{1}\{A_t\}] \lesssim \sqrt{\frac{d}{n}}\sigma_t \nu$$

Let the target matrix for all tasks be $N = \nu^2 U^* U^{*\top} + \sum_{t=1}^{T} \alpha_i \nu^2 U^* w_t w_t^\top U^{*\top}$. Define $\sigma_1 = \max_{t \in T}\{\sigma_t\}$. We can obtain the upper bound between $N$ and $\hat{N}$ using Lemma 3:

$$\mathbb{E}||\hat{N} - N||_2 \mathbf{1}\{\cap_{i=1}^{T} A_i\} \lesssim \nu^2(\frac{r}{d}\log d + \sqrt{\frac{r}{n}} + \frac{r}{n}) + \sigma_1^2(\sqrt{\frac{d}{n}} + \frac{d}{n}) + \sigma_1 \nu \sqrt{\frac{d}{n}} + \sum_{t=1}^{T}[\alpha_t \sqrt{\frac{d}{n}}\sigma_t \nu^2]$$

We can compute an upper bound on the sine distance between $U^*$ and $U_{CL}$ using Lemma 4:

$$\mathbb{E}||\sin(\Theta(U_{CL}, U^*))||_F = \mathbb{E}||\sin(\Theta(U_{CL}, U^*))||_F \mathbf{1}\{\cap_{t=1}^{T} A_t\} + \mathbb{E}||\sin(\Theta(U_{CL}, U^*))||_F \mathbf{1}\{\cup_{t=1}^{T} A_t^C\}$$

$$\lesssim \frac{\sqrt{r}\mathbb{E}||\hat{N} - N||_2 \mathbf{1}\{\cap_{t=1}^{T} A_t\}}{\lambda_r(N)} + \sqrt{r}\mathbb{P}(\mathbf{1}\{\cup_{t=1}^{T} A_t^C\})$$

$$\lesssim \frac{\sqrt{r}}{\nu^2 + \nu^2 \lambda_r(\sum_{t=1}^{\top} \alpha_t w_t w_t^\top)}(\nu^2 \frac{r}{d}\log d + \sigma_1^2 \sqrt{\frac{d}{n}} + \sum_{t=1}^{\top} \alpha_t \sqrt{\frac{d}{n}}\sigma_t \nu) + \sqrt{r}\sum_{t=1}^{T}\sqrt{\frac{d}{n}}$$

as $\alpha_t \to \infty$ under the multi-task supervised contrastive learning setting, this expression simplifies to

$$\mathbb{E}||\sin(\Theta(U_{CL}, U*))||_F \lesssim \sqrt{\frac{dr}{n}}(\sum_{t=1}^{T} \sigma_t)$$

$\square$

Next, we will use the upper bound between the contrastive learned and optimal orthonormal basis of the representations to bound the prediction risk of the downstream task.

**Theorem A 2** *Suppose $n > d \gg r$, $T > r$ and $\lambda_{(r)}(\sum_{t=1}^{T} w_t w_t^\top) > c$ for some constant $c > 0$. Let $W_{CL}$ be the learned representation using MTCon-s and $W^*$ be the optimal true representation. Then, the prediction risk of the downstream task can be bounded as:*

$$\mathbb{E}_D[\inf_{w \in \mathbb{R}^r} \mathbb{E}_E[\ell(f_{w,W_{CL}}(x), y)] - \inf_{w \in \mathbb{R}^r} \mathbb{E}_E[\ell(f_{w,W^*}(x), y)] \lesssim \sqrt{\frac{dr}{n}}(\sum_{t=1}^{T} \sigma_t)$$

**Proof:** Combining Theorem A1 with Lemma 5 gives us the desired bound,

$$\mathbb{E}_D[inf_{w \in \mathbb{R}^r} \mathbb{E}_E[\ell(f_{w,W_{CL}}(x), y)] - \inf_{w \in \mathbb{R}^r} \mathbb{E}_E[\ell(f_{w,W^*}(x), y)]] \lesssim \sqrt{\frac{dr}{n}}(\sum_{i=1}^{T} \sigma_t)$$

$\square$

## A.2 DERIVATION OF WEIGHTED MTCON LOSS

We provide a detailed derivation of the negative log-likelihood bound computation here.

$$-\log p(y|v_i^c, D, \tau, \sigma_c^2) = -\log\left(\frac{\frac{1}{|P_y^c(i)|}\sum_{p \in P_y^c(i)} \exp\left(\frac{v_i^{cT} v_p^c}{\tau \sigma_c^2}\right)}{\sum_y \frac{1}{|P_y^c(i)|}\sum_{p \in P_y^c(i)} \exp\left(\frac{v_i^{cT} v_p^c}{\tau \sigma_c^2}\right)}\right) \quad (9)$$

$$= -\log \frac{1}{|P_y^c(i)|} \sum_{p \in P_y^c(i)} \exp\left(\frac{v_i^{cT} v_p^c}{\tau \sigma_c^2}\right) + \log\left(\sum_y \frac{1}{|P_y^c(i)|} \sum_{p \in P_y^c(i)} \exp\left(\frac{v_i^{cT} v_p^c}{\tau \sigma_c^2}\right)\right) \quad (10)$$

$$\leq -\frac{1}{|P_y^c(i)|} \sum_{p \in P_y^c(i)} \log\left(\exp\left(\frac{v_i^{cT} v_p^c}{\tau \sigma_c^2}\right)\right) + \log\left(\sum_y \frac{1}{|P_y^c(i)|} \sum_{p \in P_y^c(i)} \exp\left(\frac{v_i^{cT} v_p^c}{\tau \sigma_c^2}\right)\right) \quad (11)$$

$$\approx -\frac{1}{|P_y^c(i)|} \sum_{p \in P_y^c(i)} \log\left(\exp\left(\frac{v_i^{cT} v_p^c}{\tau}\right)^{\frac{1}{\sigma_c^2}}\right) + \log\left(\sum_y \frac{1}{|P_y^c(i)|} \sum_{p \in P_y^c(i)} \exp\left(\frac{v_i^{cT} v_p^c}{\tau \sigma_c^2}\right)\right) \quad (12)$$

$$= -\frac{1}{\sigma_c^2} \frac{1}{|P_y^c(i)|} \sum_{p \in P_y^c(i)} \log\left(\exp\left(\frac{v_i^{cT} v_p^c}{\tau}\right)\right) + \log\left(\sum_y \frac{1}{|P_y^c(i)|} \sum_{p \in P_y^c(i)} \exp\left(\frac{v_i^{cT} v_p^c}{\tau \sigma_c^2}\right)\right) \quad (13)$$

$$\propto -\frac{1}{\sigma_c^2} \sum_{i=I} L_{c,i}^{mtcon} + \log\left(\frac{\sum_y \frac{1}{|P_y^c(i)|} \sum_{p \in P_y^c(i)} \exp\left(\frac{v_i^{cT} v_p^c}{\tau \sigma_c^2}\right)}{\left(\sum_y \frac{1}{|P_y^c(i)|} \sum_{p \in P_y^c(i)} \exp\left(\frac{v_i^{cT} v_p^c}{\tau}\right)\right)^{\frac{1}{\sigma_c^2}}}\right) \quad (14)$$

$$\propto -\frac{1}{\sigma_c^2} \sum_{i=I} L_{c,i}^{mtcon} + 2\log \sigma_c \quad (15)$$

We bound our negative log-likelihood with Jensen's inequality at step 2. Following the cross-entropy uncertainty log-likelihood derivation in (Kendall et al., 2018), we also use the simplifying assumption $\frac{1}{|P_y^c(i)|}\sum_{p \in P_y^c(i)} \exp\left(\frac{v_i^{cT} v_p^c}{\tau \sigma_c^2}\right) \approx \frac{1}{|P_y^c(i)|}\left(\sum_{p \in P_y^c(i)} \exp\left(\frac{v_i^{cT} v_p^c}{\tau}\right)\right)^{\frac{1}{\sigma_c^2}}$ which becomes an equality when $\sigma_c \to 1$. Extending this analysis to multi-similarity contrastive loss, we can adapt the loss function to learn weightings for each similarity. While we make several simplifying assumptions to reduce the objective, we demonstrate that the optimization objective works well in our empirical results.

## A.3 EXTENDED CLASSIFICATION EXPERIMENTS

We provide the full in-domain classification results here, including results from each of the separate single-task trained cross-entropy and supervised contrastive learning networks for the Zappos50k, CUB200-2011 and MEDIC datasets in Table 4, Table 5, and Table 6, respectively. MTCon outperforms all contrastive learning baselines with a single trained model on all tasks.

Table 4: **Extended In-domain Performance for Zappos50k.** MTCon outperforms all baselines on training tasks.

| Loss | Zappos50k | | |
| --- | --- | --- | --- |
| | Category | Closure | Gender |
| XEnt Category | 96.64 (0.34) | 74.55 (0.38) | 63.78 (0.59) |
| XEnt Closure | 88.99 (0.33) | 92.28 (0.35) | 66.59 (0.57) |
| XEnt Gender | 81.96 (0.32) | 73.28 (0.37) | 83.09 (0.60) |
| XEnt MT | 96.98 (0.29) | 93.33 (0.36) | 85.07 (0.55) |
| SimCLR | 90.05 (0.43) | 81.30 (0.49) | 69.10 (0.84) |
| SupCon Category | 96.95 (0.29) | 73.02 (0.36) | 61.24 (0.62) |
| SupCon Closure | 83.62 (0.30) | 91.75 (0.41) | 65.90 (0.60) |
| SupCon Gender | 76.40 (0.28) | 69.52 (0.38) | 85.11 (0.58) |
| CSN | 83.33 (0.32) | 72.12 (0.36) | 69.21 (0.60) |
| SCE-Net | 86.23 (0.31) | 75.32 (0.33) | 71.32 (0.59) |
| MTCon | **97.17 (0.27)** | **94.37 (0.35)** | **85.98 (0.56)** |

## A.4 SIMILARITY SPACE ANALYSIS

**Zappos50k** We visually examine the embeddings learned through our approach to qualitatively analyze the learned similarity projection spaces. Figure 4 shows the 2D t-SNE (Van der Maaten

Table 5: **Extended In-domain Performance for CUB200-2011.** MTCon outperforms all baselines on training tasks.

| Loss | CUB200-2011 | | |
|---|---|---|---|
| | Shape | Size | Primary Color |
| XEnt Shape | 55.76 (0.50) | 52.41 (0.47) | 26.44 (0.44) |
| XEnt Size | 54.54 (0.46) | 55.91 (0.48) | 22.02 (0.40) |
| XEnt Primary Color | 54.57 (0.49) | 55.96 (0.47) | 32.61 (0.45) |
| XEnt MT | 54.87 (0.49) | 56.96 (0.47) | 33.18 (0.45) |
| SimCLR | 34.20 (0.46) | 52.43 (0.48) | 28.51 (0.43) |
| SupCon Shape | 55.92 (0.49) | 58.16 (0.47) | 31.82 (0.45) |
| SupCon Size | 54.44 (0.50) | 58.13 (0.48) | 30.99 (0.46) |
| SupCon Primary Color | 54.79 (0.48) | 53.41 (0.48) | 33.28 (0.47) |
| CSN | 45.14 (0.49) | 48.24 (0.45) | 25.23 (0.42) |
| SCE-Net | 48.29 (0.41) | 51.53 (0.44) | 28.78 (0.41) |
| MTCon | **56.88 (0.49)** | **59.32 (0.48)** | **35.97 (0.45)** |

Table 6: **Extended In-domain Performance for MEDIC.** MTCon outperforms all contrastive learning baselines on training tasks.

| Loss | MEDIC: In-Domain Evaluation | | | |
|---|---|---|---|---|
| | Damage severity | Disaster types | Humanitarian | Informative |
| XEnt Damage severity | **81.39 (0.35)** | 75.71 (0.37) | 81.76 (0.33) | 84.48 (0.31) |
| XEnt Disaster types | 81.02 (0.34) | 78.98 (0.35) | 82.06 (0.34) | 86.08 (0.3) |
| XEnt Humanitarian | 81.32 (0.36) | 76.52 (0.35) | **82.1 (0.37)** | **86.41 (0.31)** |
| XEnt Informative | 80.2 (0.36) | 76.73 (0.35) | 80.83 (0.36) | 85.68 (0.3) |
| XEnt Multi-Task | 81.01 (0.36) | 78.04 (0.32) | **82.25 (0.35)** | 86.01 (0.29) |
| SimCLR | 74.9 (0.4) | 68.5 (0.42) | 73.89 (0.4) | 78.67 (0.33) |
| SupCon Damage severity | 80.26 (0.33) | 75.1 (0.4) | 80.42 (0.4) | 84.45 (0.34) |
| SupCon Disaster types | 80.23 (0.34) | 78.33 (0.37) | 80.63 (0.36) | 84.02 (0.3) |
| SupCon Humanitarian | 79.98 (0.36) | 74.89 (0.39) | 80.36 (0.32) | 85.07 (0.32) |
| SupCon Informative | 79.14 (0.35) | 74.67 (0.34) | 79.97 (0.31) | 84.02 (0.3) |
| CSN | 75.13 (0.4) | 70.02 (0.37) | 70.52 (0.38) | 76.28 (0.32) |
| SCE-Net | 77.25 (0.42) | 71.15 (0.39) | 72.12 (0.42) | 77.52 (0.33) |
| MTCon | 81.0 (0.3) | **79.14 (0.31)** | 81.69 (0.3) | 85.15 (0.3) |

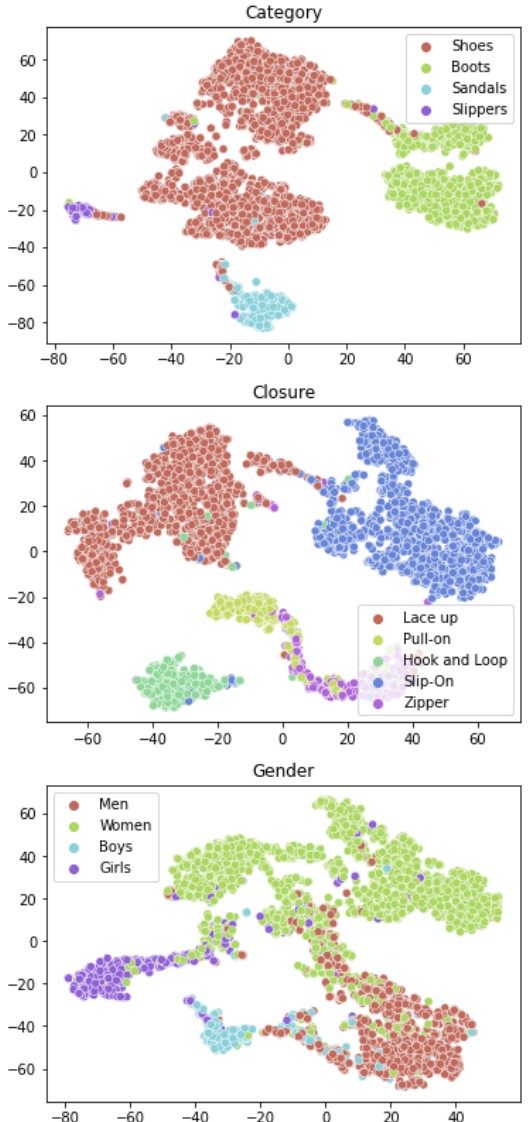

Figure 4: **2D T-SNE plots of projection spaces learned through our approach (MTCon) trained on the Zappos50k dataset.** Top, middle and bottom plots show category, closure, and gender projection spaces respectively. Our approach correctly learns to separate disjoint similarity metrics.

& Hinton, 2008) plots for each of the category, closure, and gender similarity metrics represented in the Zappos50k test dataset. Figure 4 indicates that MTCon correctly learns to separate these disjoint similarity metrics. Surprisingly, the figure also suggests that MTCon is able to learn some inherent structure of the underlying data that is not explicitly represented in the similarity labels. For example, in the gender projection space, MTCon appears to learn that women's shoes are more similar to both men's and girl's shoes than to boy's shoes. We present a similar analysis on the MEDIC dataset in the Appendix, showing that the same conclusions hold.

**MEDIC** We visually demonstrate that the projection spaces trained with multi-similarity contrastive loss are learning to separate disjoint similarity metrics. Figure 5 shows the 2D t-SNE plots for each of the damage severity, disaster types, humanitarian, and informative similarity metrics represented in the MEDIC test dataset. Figure 5 indicates that the multi-similarity contrastive network is learning how to separate these disjoint similarity metrics. It also suggests that these projected subspace learn some inherent structure of the underlying data that is not explicitly represented in the

similarity metric. For example, in the disaster types projection space, it appears that landslides are more similar to earthquakes than to fires. Similarly, hurricanes and floods appear relatively similar in the space. We also notice a gradient of damage severity, where the mild damage severity images typically appear in between severe and little or none damage severity images.

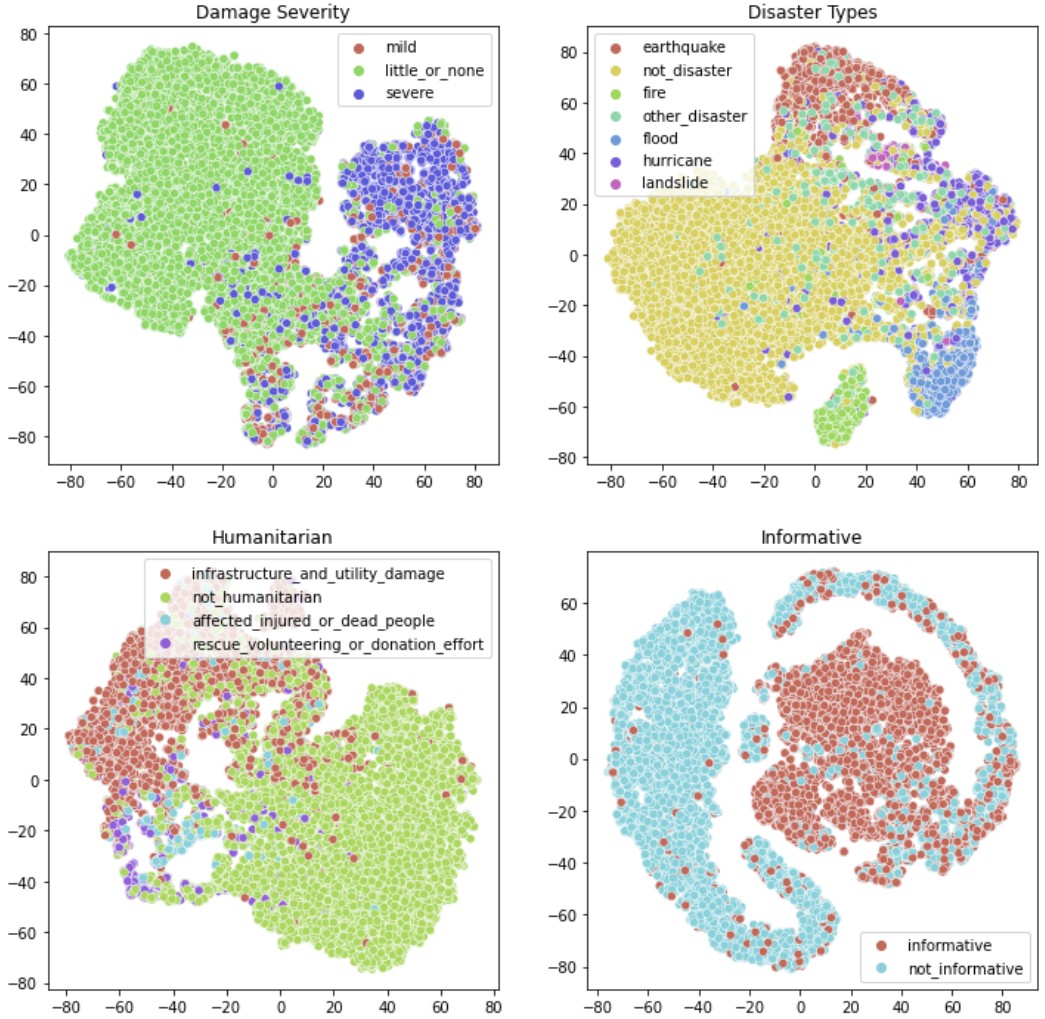

Figure 5: **T-SNE plots for each of the projection spaces of a multi-similarity contrastive network trained on the MEDIC dataset.** The four plots show the 2D T-SNE plot for the category, closure, and gender projection spaces of the MEDIC test set respectively.

## A.5  HYPERPARAMETER ANALYSIS.

We test if there exists a specific temperature that leads to optimal performance of MTCon for multiple similarity metrics. In Figure 6, we plot the top-1 classification accuracy for each of the category, closure, and gender tasks as a function of pretraining temperature for MTCon. We also plot the top-1 classification accuracy as a function of training epochs. We find that a pretraining temperature of $\tau = 0.1$ and training for 200 epochs works well for all tasks. These hyperparameter settings are consistent with optimal hyperparameter settings for SimCLR and SupCon. Note that previous work for SimCLR and SupCon have found the large batch sizes consistently result in better top-1 accuracy (Chen et al., 2020; Khosla et al., 2020). We hypothesize that larger batch sizes would also improve performance for MTCon loss. We include hyperparameter analyses on MTCon for the MEDIC dataset in Figure 7.

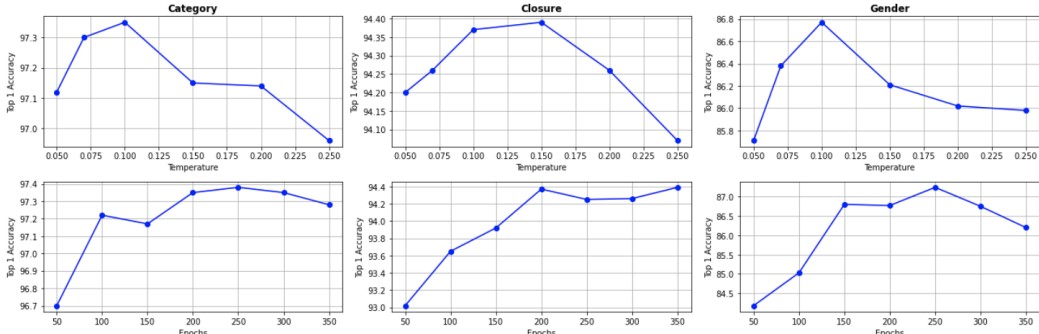

Figure 6: **Hyperparameter effect on top-1 accuracy for the Zappos50k dataset.** The top row shows top-1 classification accuracy as a function of temperature during pretraining stage for MTCon. The bottom row shows top-1 classification accuracy as a function of pretraining epochs for MTCon.

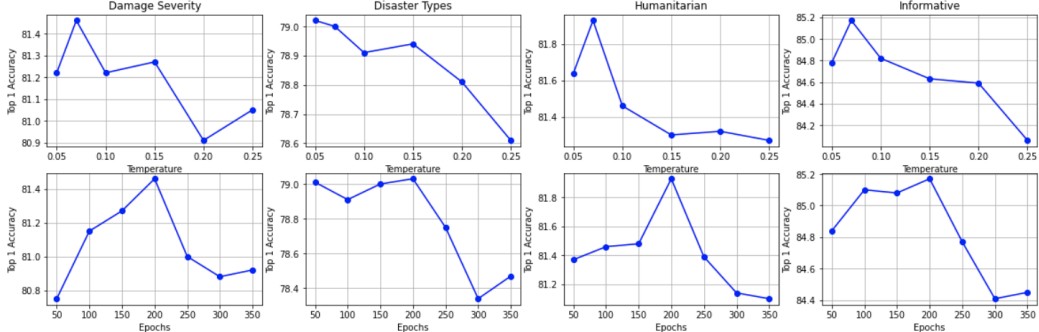

Figure 7: **Hyperparameter effect on top-1 accuracy for the MEDIC dataset.** The top row shows top-1 classification accuracy as a function of temperature during pretraining stage for MTCon. The bottom row shows top-1 classification accuracy as a function of pretraining epochs for MTCon.

