# OpenReview forum: "Multitask Contrastive Learning"
_ICLR.cc/2024/Conference — Submitted to ICLR 2024_

### Official Review · Reviewer_TwpG · 2023-11-01

**Soundness:** 3 good
**Presentation:** 2 fair
**Contribution:** 2 fair
**Rating:** 5
**Confidence:** 4

**Summary:**

**Summarization**:
This paper proposes a multi-task contrastive loss (MTCon) to combine the multi-task domain and contrastive learning. The main contributions of this paper could be summarized into one point, i.e., incorporating task weightings that consider the uncertainty of each task, reducing the impact of uncertain tasks, and leading to better out-of-domain generalization for unseen tasks.

**Reasons To Accept**:
1. A multi-task contrastive loss. The paper introduces a multi-task contrastive loss MTCon, which combines contrastive learning with multi-task scenarios. This loss showcases the potential for enhanced embedding generalization across tasks.
2. A weighting scheme. This paper incorporates task weighting, offering a mechanism to address uncertainties in different tasks.

**Reasons To Reject**:
1. Unclear Motivation: The paper lacks a clear and convincing motivation, especially considering the abundance of prior work that combines multi-task and contrastive learning [1-3]. The authors do not provide a coherent rationale for this combination, and the specific domain and problem targeted remain unclear.

2. Insufficient Novelty: The proposed method appears overly simplistic and lacks significant innovation. From the outlined approach in the paper, it appears to have limitations in terms of generalizability and transferability, with limited performance across different datasets.

3. Failure to Address Potential Conflicts Among Different Similarity Notions: Multi-task learning often involves different similarity metrics, and the paper does not seem to consider how to handle potential conflicts or issues arising from the use of multiple similarity notions.

4. Outdated Comparative Methods: The chosen comparative methods in the paper do not appear to represent the state-of-the-art in the field. The paper lacks sufficient evidence to demonstrate the competitiveness of the proposed approach within the competitive research landscape.

5. Lack of Targeted Title and Unclear Pipeline: The title should ideally provide a clear indication of the paper's focus and contributions, while the pipeline should serve as a visual guide for readers to understand the proposed methodology. The absence of a targeted title and an unclear flowchart can hinder the paper's accessibility and understanding, making it challenging for readers and researchers to grasp the core message and methodology.

**Summary Of the Review**:

In summary, this paper introduces a multi-task contrastive loss (MTCon) that combines multi-task scenarios and contrastive learning, primarily by incorporating task weightings to address uncertainty in tasks and improve out-of-domain generalization. While the contributions are promising, the paper suffers from unclear motivation, limited novelty, a lack of consideration for handling similarity conflicts, and outdated comparative methods. Furthermore, the title of the paper lacks specificity, and the absence of a clear flowchart hinders accessibility and understanding. These combined factors indicate that the paper does not meet the standard for acceptance.


Reference:
[1] Ravikiran Parameshwara, Ibrahim Radwan, Akshay Asthana, Iman Abbasnejad, Ramanathan Subramanian, Roland Goecke: Efficient Labelling of Affective Video Datasets via Few-Shot & Multi-Task Contrastive Learning. ACM Multimedia 2023: 6161-6170
[2] Junichiro Iwasawa, Yuichiro Hirano, Yohei Sugawara: Label-Efficient Multi-task Segmentation Using Contrastive Learning. BrainLes@MICCAI (1) 2020: 101-110
[3] Yu Zhang, Hao Cheng, Zhihong Shen, Xiaodong Liu, Ye-Yi Wang, Jianfeng Gao:
Pre-training Multi-task Contrastive Learning Models for Scientific Literature Understanding. CoRR abs/2305.14232 (2023)

**Strengths:**

See summary

**Weaknesses:**

See summary

**Questions:**

See summary

---

> ### Author Response · Authors · 2023-11-14
>
> Rebuttal Response: We thank the reviewer for their careful review and feedback. We address the individual weaknesses raised below:
>
> 1. We agree with the reviewer that bespoke multi-task contrastive learning methods have been developed under specific domains. In comparison to the works cited by the reviewer (which focuses on domain specific tasks such as segmentation) and additional works in the hierarchical contrastive learning space (Use all the Labels, Zhang et al. 2022), we focus on the general multi-task classification space where multiple tasks can have disjoint labels. Specifically, none of the domain specific papers handle the case where two images, e.g., of shoes, can be labeled as positive examples under one task but negative examples under another. However, we believe that the papers the reviewer cited would add to the depth of the paper, and we will add them to the relevant works section.
> 2. We provide theoretical and empirical evidence that weighting uncertainty for source tasks can improve out-of-domain generalization and find that MT-Con outperforms the second-best weighted multi-task baseline by an average of over 1.5% across out-of-domain tasks (Table 1).
> 3. The point that the reviewer brings up is very interesting. Addressing conflicting similarity metrics is one of the core contributions of our work, setting it apart from the bespoke approaches that the reviewer highlighted. In Figure 1, we highlight what happens in the scenario that the reviewer raises, where the similarity metrics are contradicting. The construction of the network with multiple similarity heads enables MT-Con to handle conflicting similarity metrics. We will highlight this better in a revision.
> 4. We compare to the most recent general multi-similarity learning methods, including SCE-Net and CSN [used as upper baselines in 1, 2]. We also compare to the most popular weighted general multi-task cross-entropy method (Kendall et al. 2018). Could the reviewer please point us to any general multi-similarity (can handle conflicting metrics) methods they would like to see?
> 5. We tried to capture the multiple disjoint similarity metrics with Figure 1 as motivation. Could the reviewer please clarify a suggested title or pipeline that would help with general understanding?
>
> [1] “How Good Is Aesthetic Ability of a Fashion Model?” (Zou et al. 2022)
> [2] “Identifying ambiguous similarity conditions via semantic matching” (Ye et al. 2022)

---

### Official Review · Reviewer_BR7u · 2023-11-03

**Soundness:** 3 good
**Presentation:** 3 good
**Contribution:** 1 poor
**Rating:** 5
**Confidence:** 3

**Summary:**

This paper introduces a multi-task contrastive learning loss function, MtCon, which learns representations under many measures of similarity between examples. They show strong results on three multi-task vision datasets over vanilla contrastive learning baselines.

**Strengths:**

- The paper is clearly written and easy to follow.
 - The method (MtCon) is simple, and the authors provide an uncertainty-motivated derivation of their method.
 - The experiments on the three datasets (Zapp050k, CUB200-2001, MEDIC) show that MtCon works better in multi-task settings compared to the chosen baselines.

**Weaknesses:**

- The authors provide a derivation of the MtCon loss function to arrive at Eqn (8), but it ends up simply learning a scalar weight for different contrastive learning tasks and regularizing the scalar weights. Seeing that it is so straightforward, I think the paper culd benefit from 1. more discussion on if this loss function is better suited for *specifically* for contrastive learning (unless I'm missing something, most of the analysis and theory could apply to any multi-task setting, even if it isn't contrastive), and 2. more task-weighting baselines like the ones mentioned in prior work. The one multi-task weighting baseline XEnt-MT already seems pretty close to MtCon in terms of performance, so I imagine the other multi-task weighting methods work well too.
 - Even at high noise levels, the weight of the noisy tasks doesn't fall that low compared to other tasks (in Figure 3). I think a useful ablation might be to manually set the weight of the noise task to a very low or very high number and show how performance changes.

**Questions:**

- How is the MtCon weighting scheme specifically suited for contrastive learning? If it isn't specific to contrastive learning, should more multi-task weighting baselines be compared to?
  * Is this method just a simple multi-task weighting scheme applied to contrastive learning tasks?
 - How does the MtCon weighting scheme connect to prior work on multi-task weighting?

I'm open to having my mind changed, looking forward to your response.

---

> ### Author Response · Authors · 2023-11-14
>
> Rebuttal Response: We thank the reviewer for their careful review and feedback and appreciate the reviewer’s openness to further discussion. We address the individual weaknesses and questions raised below:
>
> Weaknesses:
> 1. The idea behind MTCon weighting is to downweight tasks with more task uncertainty. As the reviewer correctly points out, this idea has been previously explored in the multi-task literature, but not in the contrastive setting.  Existing techniques do not extend directly to the contrastive setting because unlike existing methods, contrastive loss doesn't imply a likelihood function over the observed data. Thus, we develop the pseudo-likelihood to demonstrate that the temperature corresponds to uncertainty weighting. We do find improvements to using contrastive learning in conjunction with the multi-task learning (similar to previous self-supervised and supervised contrastive learning work such as Khosla et al 2020). Across out-of-domain tasks, MT-Con outperforms uncertainty weighted multi-task cross-entropy by an average of over 1.5% across out-of-domain tasks (Table 1).
> 2. This is a great question.. As the uncertainty increases, the learned weight should converge to zero, but does not because of estimation error. We expect such estimation error to go to zero as the sample size goes to infinity. We like the idea of studying the behavior in a regime where we have no estimation error. When we manually set the task weight to 0 for the corrupted tasks, the generalization accuracies for the Zappos50k and MEDIC datasets are 40.5% and 80.8%, respectively. We will include these results in the Appendix.

---

### Official Review · Reviewer_EEgJ · 2023-11-04

**Soundness:** 3 good
**Presentation:** 3 good
**Contribution:** 3 good
**Rating:** 6
**Confidence:** 4

**Summary:**

The paper aims to develop a new method, Multi-Task Contrastive Loss (MTCon), which combines contrastive learning and multi-task learning to obtain robust representations that capture multiple similarity metrics. MTCon achieves this by learning task weights that reflect the uncertainty associated with tasks. Experimental results on three multi-task datasets, Zappos50k, MEDIC and CUB200- 2011, show that the proposed approach enhances generalization performance on out-of-domain tasks. Furthermore, the proposed approach  has better performance than the weighted multi-task cross-entropy counterpart for both in-domain and out-of-domain scenarios.

**Strengths:**

The paper introduces a novel result that combines multi-task learning and contrastive learning.

Theoretical results are proven for a simplified version of the problem.

Experimental results on three datasets show that the proposed approach based on the multi-task contrastive loss has overall better results than a similar model based on the cross-entropy loss.

**Weaknesses:**

The results on the MEDIC set do not support the overall claims and conclusions of the study. Specifically, the paper claims that the proposed model learns task weights that capture the uncertainty of the tasks. However, when the results on some of the MEDIC tasks are inferior as compared to those of the baselines, the authors speculate that is due to higher uncertainty in those tasks. This seems to be a circular argument as the main assumption of the proposed approach is that the task uncertainty can be learned through task-specific weights. More analysis of those tasks and their weights is needed to better understand when the proposed approach helps and when it may not.

**Questions:**

The authors cite  Alam et al. (2018; 2022) to explain the negative results for some of the MEDIC tasks.  But what was Alam et al.'s  basis for concluding that there is  inherent uncertainty for some of the MEDIC  tasks? it would be interesting to know how strong their argument was to avoid propagating information that may only be speculative.

---

> ### Author Response · Authors · 2023-11-14
>
> Rebuttal Response: We thank the reviewer for their careful review and feedback. We appreciate that the reviewer recognized the theoretical and empirical contributions of MTCon. We address the individual weaknesses and questions raised below:
>
> Weaknesses & Questions:
> We thank the reviewer for the thoughtful comments on the work. As the reviewer observes, our method performs comparably to others on MEDIC **in-domain** tasks, but MTCon does generalize better than baselines on **out-of-domain** tasks (Table 1), which is our main goal in this work. We will highlight this nuance more clearly in the final version.  We direct the reviewer to Figure 3, the ablation study. We find that weighted MTCon learns weightings that are robust to individual task noise in order to generalize better to out-of-domain tasks even when in-domain performance remains similar.
>
> Alam et al., Table 10, shows that adding the informativeness task to multi-task training actually decreases performance for the other tasks and will improve the main text correspondingly.

---

> > ### Comment · Reviewer_EEgJ · 2023-11-17
> >
> > I have read the rebuttal. I will not increase my score, as my concerns still hold.

---

### Official Review · Reviewer_yq2P · 2023-11-05

**Soundness:** 2 fair
**Presentation:** 2 fair
**Contribution:** 2 fair
**Rating:** 5
**Confidence:** 4

**Summary:**

This paper introduces Multi-Task Contrastive Loss (MTCon), a new method that combines multi-task and contrastive learning to improve representation learning. MTCon uses multiple projection heads to handle different notions of similarity across tasks. It also incorporates a weighting scheme to downweight more uncertain tasks, improving generalization. Through experiments on 3 datasets, MTCon is shown to outperform multi-task cross-entropy and prior contrastive methods on both in-domain and out-of-domain tasks. For example, it improves average out-of-domain accuracy by 3.3% over multi-task cross-entropy. Analysis indicates the weighting scheme helps MTCon better handle noise in the training tasks. The paper also provides theoretical analysis bounding generalization error based on task noise levels. Overall, MTCon introduces a novel approach to multi-task contrastive learning that achieves state-of-the-art performance by handling multiple similarity metrics and task uncertainty.

**Strengths:**

- The paper presents a novel approach for combining multi-task and contrastive learning, which to my knowledge has not been done before. The use of multiple projection heads and learned weighting scheme specifically for handling multiple disjoint similarity metrics is creative and original.
- The paper is well-organized and clearly explains both the proposed method and experiments. The problem formulation and notation are clear.
- This work makes both empirical and theoretical contributions. It pushes forward the state-of-the-art in representation learning, achieving superior performance to prior multi-task and contrastive methods.

**Weaknesses:**

- The theoretical analysis makes some simplifying assumptions (e.g. abundance of source tasks) that may not perfectly hold in practice.
- All datasets used are for computer vision. Testing MTCon on a wider variety of modalities (text, audio, etc) could better demonstrate generalization.
- The experimental evaluation is quite thorough, but lacks ablation studies to isolate the impact of different components of MTCon (e.g. projection heads vs weighting scheme). Ablation studies would provide more insight.
- The comparison to prior work is limited to a few baselines. Comparing against a broader range of multi-task representation learning methods could better situate MTCon.
- The hyperparameter analysis is quite brief. A more extensive sweep over training hyperparameters and architectural choices could be illuminating.

**Questions:**

- Have you considered any other proxies for estimating task uncertainty besides the constructed pseudo-likelihood? How does using other uncertainty estimates impact performance?

---

> ### Author Response · Authors · 2023-11-14
>
> Rebuttal Response: We thank the reviewer for their careful review and feedback. We appreciate that the reviewer recognized the theoretical and empirical contributions of MTCon. We address the individual weaknesses and questions raised below:
>
> Weaknesses:
> 1.  The reviewer is correct in that our theoretical analysis assumes that the source tasks contain sufficient information to cover target tasks. We stress, however, that this assumption is in line with the majority of theoretical work in the multitask and supervised contrastive learning fields, including [1] and [2]. The experimental results are consistent with what the theory suggests.
> 2. It would indeed be an interesting future direction to apply our method to other modalities for which multiple similarity measures are common, but this work focuses on showing that the methods work for images on three multi-task vision datasets from very different domains. These datasets are a superset of those used in earlier contrastive multi-similarity representation work, including [3] and [4].
> 3. We provide an ablation study in Figure 3 that demonstrates that weighted MT-Con generalizes better than the unweighted version to out-of-domain tasks. We cannot ablate the multiple projection heads since MT-Con is dependent on it to capture disjoint similarities.  Can the reviewer clarify if there are other specific ablations they believe would strengthen the paper?
> 4. We provide comparisons to self-supervised, single-task supervised, and the most recent multi-task multi-similarity supervised contrastive learning methods across all three evaluation datasets. We chose these baselines because they are commonly used in the literature [5,6,7].  We would be happy to include other baselines that the reviewer would like to see.
> 5. We provide hyperparameter analyses for temperature and number of epochs for both the Zappos50k and MEDIC datasets in Figures 6,7 in the Appendix. We can change the main text to summarize these.
>
> Questions:
> 1. We thank the reviewer for the thoughtful question. We have not considered other metrics of uncertainty since we found that the pseudo-likelihood worked well for us in theory and in practice. We believe that this would be an interesting direction for future work.
>
> [1] “The power of contrast for feature learning: A theoretical analysis” (Ji et al. 2021)
> [2] “Few-shot learning via learning the representation, provably” (Du et al. 2020)
> [3] “Conditional similarity networks” (Veit et al. 2017)
> [4] “SCE-Net” (Tan et al. 2019)
> [5] “How Good Is Aesthetic Ability of a Fashion Model?” (Zou et al. 2022)
> [6] “Identifying ambiguous similarity conditions via semantic matching” (Ye et al. 2022)
> [7] “Multi-task learning using uncertainty to weigh losses for scene geometry and semantics” (Kendall et al. 2017)

---

### Meta-Review · Area_Chair_UBxL · 2023-12-08

**Metareview:**

There are many concerns raised by the reviewers in their original concerns including 1, the assumptions made for the theoretical study in this work hardly hold in practice; 2, as the proposed method is a general ML approach, it would be more convincing to conduct experiments not only on CV datasets; 3, a more comprehensive ablation study is required to further understand the impacts of different components and hyper-parameters of the proposed method; 4, the technical novelty of the proposed method is limited.  During rebuttal, the authors fail to address the major concerns. Therefore, this work is not ready for publication.

**Justification For Why Not Higher Score:**

Major concerns have not ben addressed.

**Justification For Why Not Lower Score:**

N/A

---

### Decision · Program_Chairs · 2024-01-16

Reject